# Coping Strategies and Personality Profile Characteristics of People Whose Parents Were Alcohol Addicts

**DOI:** 10.3390/bs10010032

**Published:** 2020-01-10

**Authors:** Anna M. Lutsenko

**Affiliations:** Department of Abnormal Psychology, Faculty of Psychology, Lomonosov Moscow State University, Mochovaya 9, 125009 Moscow, Russia; lutscenko.anna@mail.ru; Tel.: +7-915-690-77-73

**Keywords:** twelve-step recovery program, alcohol addiction, dysfunctional family, family pain, guilt, coping strategies

## Abstract

The relevance of this research is due to the wide prevalence of addictive behavior and the insufficient knowledge of the coping strategies of patients and their families. The purpose of this research was to investigate the resource factors and coping strategies of adults with alcohol-addicted parents and to make recommendations for psychological counseling for these people. The sample consisted of 52 subjects—who were participants in a twelve-step rehabilitation program for adult people whose parents had alcohol addictions—and 50 controls. We used guilt questionnaires (“The Guilt Inventory Questionnaire”, “The Interpersonal Guilt Questionnaire”), quantitative methods for evaluating the coping strategies used by participants (“coping strategies” (Lazarus, Folkman)), and a phenomenological analysis of the interviews with the participants. The results showed that adults with alcohol-addicted parents felt guilty in situations when they took care of somebody because their own parents did not model (and teach them) caretaking behavior. People whose parents were alcohol addicts tend to avoid accepting responsibility for their lives. The resource factors of people with alcohol-addicted parents included keeping a diary, participation in a rehabilitation program, and confidential communication with other people.

## 1. Introduction

The problem of addictive behavior has been actively studied in the last five years in Russia and in Europe [1,2,3,4,5,6,7,8,9,10]. The popularity of this topic is associated with the very wide prevalence of this disease. According to the Ministry of Health of the Russian Federation, in Russia, there were over 12 million people with alcohol-addicted parents in 2017. Most of the psychological studies of alcohol addiction are focused on the personality of the patient—the study of their cognitive and emotional features [7,9,11]. However, the characteristics of the patient’s family system, the distribution of family roles, the family rules, and the negative emotions experienced by family members can play an important role in the development of alcohol addiction and in the effectiveness of its medical and psychotherapeutic treatment [5,12]. 

In family system psychotherapy, the family is considered as a social system consisting of a set of elements (the family members and all interactions between them) that exist in dynamic relationships with each other [13]. The life of the family system conforms to two laws: Homeostasis and development. A dysfunctional family is a family where the effect of the law of homeostasis is stronger than the effect of the law of development. A family with an alcohol addict is certainly a dysfunctional family. In such families, according to the research of V.D. Moskalenko, family communication is disturbed, unrealistic family rules appear in family communication, and family roles and functions are transformed [11]. Family psychotherapists often use the term “the family pain” to refer to the emotional state of the dysfunctional family members [13]. Virginia Satir identified two types of family pain: A condition resulting from an awareness of family problems and the feeling of guilt due to family events [14]. The feeling of guilt has been studied less than primary emotions because it cannot be directly observed. In psychoanalysis, guilt is the result of tension between the achievements of the ego and the demands of the superego. In existentialism, it appears when a person realizes that s/he has responsibility for his/her own being. In cognitive psychology, the guilt is investigated as a result of attributing to oneself the causality of events [15]. In this study, the feeling of guilt is considered as “a painful state of awareness that accompanies the accomplished or intentional violation of social values and rules”.

J. Woititz studied people whose parents were alcohol addicts and distinguished their common features [12]. These people do not have the model of caretaking family behavior, they are afraid of everything new in their life, they report low self-esteem, and they frequently experience guilt and difficulty in establishing intimate relationships in their own families. In the Russian studies conducted by A.V. Kuritsyn and A.V. Merinov, children in families of alcoholics tended to be co-dependent, they demonstrated addictive and suicidal behavior, they reported low self-esteem, and they experienced difficulties in establishing close relationships with other people [2,5,6]. Toney A. was the founder of the twelve-step rehabilitation program for adults whose parents were chemical addicts. This program includes attending open and closed meetings, being prescribed 12 steps under the guidance of a mentor, and preparing presentations at meetings [16]. Several recent empirical and theoretical studies show that people with alcohol-addicted parents usually feel guilty, but these investigations were conducted either on people who suffered from chemical addictions or on mentally ill individuals [1,5,8,12]. In the current study, the experimental group consisted of mentally healthy adults who did not suffer from chemical addictions but whose parents suffered from alcohol addiction. The control group consisted of mentally healthy adults who did not suffer from addictions and whose parents were not alcohol addicts. 

The purpose of this research was to investigate the coping strategies of people whose parents had alcohol addiction and to make recommendations for psychological counseling for these people. We suggested that (1) there are significant differences between the coping strategies used by adult people from healthy families versus adult people whose parents suffered from alcohol addiction, (2) the feeling of guilt experienced by people with alcohol-addicted parents is associated with the actual condition of their parents, and (3) the coping strategies can be transformed during the process of participation in the twelve-step rehabilitation program “Adult Children of Alcoholics”. 

## 2. Methods

### 2.1. Participants

The study was conducted in the period from October 2018 to March 2019. The sample consisted of 52 subjects (11 men and 41 women; mean age = 24.5)—they were participants in a twelve-step rehabilitation program for adult people whose parents suffered from alcohol addiction—and 50 controls (15 men and 35 women; mean age = 24.2). The inclusion criteria required all participants to range from 18 to 35 years of age and to report being mentally healthy adults who were not alcohol addicts. People whose parents suffered from alcohol addiction were recruited in the twelve-step rehabilitation program “Adult Children of Alcoholics” [16]. One of their parents suffered from alcohol addiction and was treated for the disease throughout his/her life. Alcohol addiction occurred during the period when the participants of this study were from 6 months to 5 years old, and parents were directly involved in the upbringing of their children. Controls were recruited through social networks; the researcher talked with the participants about their families before including them in the sample. 

All subjects gave their informed consent for inclusion before they participated in the study. The study was conducted in accordance with the Declaration of Helsinki, and the protocol was approved by the Ethics Committee of Lomonosov Moscow State University (№38, 18.10.2017). The participants did not receive any material rewards for their participation. All participants were given feedback on the results of this study with indications of the resource factors of the patients’ family systems. 

### 2.2. Measures

We used guilt questionnaires the “The Guilt Inventory Questionnaire” and “The Interpersonal Guilt Questionnaire”, quantitative methods for evaluating the coping strategies used by participants (“Coping strategies” (Lazarus, Folkman)), and a phenomenological analysis of the interviews with the participants.

“The Guilt Inventory Questionnaire” (Kugler, Jones) consists of 45 statements [17]. The results for the guilt scale primarily pertained to three subscales: The state of guilt, the feeling of guilt, and moral norms. The “state of guilt” scale indicates the temporary emotional state that a person is currently experiencing. The “feeling of guilt” scale indicates stable personality characteristics. The “moral norms” scale indicates the tendency to observe moral standards. “The Interpersonal Guilt Questionnaire” (O’Connor, Berry) consists of 45 statements [18]. The results for the guilt scale primarily pertained to four subscales: The survivor guilt, the guilt of separation, the guilt of responsibility, and the feeling of shame. “The coping strategies” (Lazarus, Folkman) consists of 50 statements. The results for the coping strategies pertained to 8 subscales. 

The phenomenological analysis of the interviews with the participants consisted of collecting information (participants’ statements about their feelings and resource factors), identifying and transforming the semantic units of the participants’ statements, and grouping the semantic units by topics and interpreting the topics received [19]. The information mentioned in this study is printed with the permission of the participants. Three participants were excluded from this study because it turned out that they were drug addicts.

### 2.3. Statistical Methods 

All quantitative analyses were conducted using SPSS statistics (Version 22.0). We used the non-parametric Mann–Whitney test and the parametric T-test for independent variables to determine differences between the samples. We also calculated the effect size (Cohen’s d) indicating the difference between the samples for each comparison. Effect sizes (Cohen’s d) of 0.20 can be described as small effects, effect sizes of 0.50 as medium effects, and effect sizes of 0.80 as large effects. 

## 3. Results

### 3.1. Coping Strategies and Guilt Questionnaires

The results of “The Guilt Inventory Questionnaire” (Kugler, Jones) showed that the feeling of guilt experienced by people with alcohol addict parents (ex.) scored significantly higher than the feelings of guilt in people from the control group (cont.) (M cont. = 56.58, SD (standard deviation) cont. = 14.69; M ex. = 63.90, SD ex. = 12.63; *t* = 2.703; *p* = 0.008; *d* = 0.535), as seen in Table 1 and Table 2. However, there were not significant differences between the states of guilt experienced by adult people from healthy families versus adult people whose parents suffered from alcohol addiction (M cont. = 27.64, SD cont. = 7.98; M ex. = 27.96, SD ex. = 7.58; *t* = 0.209; *p* = 0.835; *d* = 0.041). 

The results of “The Interpersonal Guilt Questionnaire” (O’Connor, Berry) showed that the guilt of responsibility experienced by people whose parents had alcohol addictions scored significantly higher than the guilt of responsibility experienced by people from the control group (M cont. = 61.48, SD cont. = 12.95; M ex. = 68.81, SD ex. = 13.43; *t* = 2.804; *p* = 0.006; *d* = 0.556). The separation guilt experienced by adults whose parents were alcohol addicts scored significantly lower than the separation guilt experienced by people from the control group (M cont. = 38.46, SD cont. = 9.40; M ex. = 34.44, SD ex. = 9.57; *t* = 2.137; *p* = 0.035; *d* = 0.423). There were not significant differences between the feeling of shame experienced by adult people from healthy families versus adult people whose parents were alcohol addicts (M cont. = 40.04, SD cont. = 13.72; M ex. = 44.40, SD ex. = 13.76; *t* = 1.604; *p* = 0.112; *d* = 0.318). 

People whose parents were alcohol addicts used escape–avoidance as coping strategies more often (M cont. = 10.72, SD cont. = 3.28; M ex. = 14.08, SD ex. = 4.05; *t* = 4.586; *p* = 0.000; *d* = 0.908) than people from healthy families.

### 3.2. Phenomenological Analysis of the Interviews

The phenomenological analysis of the interviews with the participants consisted of collecting information (participants’ statements about their feelings), identifying and transforming the semantic units of the participants’ statements, grouping the semantic units by topics, and interpreting the topics received. Five categories related to experiencing the feeling of guilt were identified during the analysis (Appendix A).

1. “The guilt of responsibility”: This type of guilt emerged in situations where the participants of this study took increased responsibility for other people and forgot about their own needs. A total of 48 program members mentioned this type of guilt.

2. “Guilt as a taboo offense against close relatives”: Only three program members mentioned this type of guilt. Mentioning the feeling of guilt towards parents, the participants also spoke about the anger that they felt towards their parents or how they showed this anger in their behavior (increased voice, mention of aggression in their life).

3. “Guilt as a fear of death”: The parents of 32 study participants died at the time of this study. However, the feeling of guilt as a fear of the death of close relatives was mentioned by the participants whose parents died and by the participants who are experiencing that their parents will die from alcoholism in future.

4. “Guilt as insufficient attention to themselves”: A total of 34 program members mentioned this type of guilt. This feeling emerged in situations where the participants of the recovery program recognized that they did not pay enough attention to their needs because they thought about their parents.

5. “The feeling of guilt as a permanent condition”: A total of 50 program members mentioned this type of guilt. The feelings of guilt were described by the participants as a permanent state that has been accompanying them since their childhood.

### 3.3. A Phenomenological Analysis of the Transcripts of the Public Meetings of the Twelve-Step Rehabilitation Program

A phenomenological analysis showed that there are three main stages of overcoming the negative feelings experienced by people whose parents suffered from alcohol addiction: awareness of feelings, studying of the characteristics of the feelings and their impact on their families, and the search for the coping strategies of the patient’s family system. 

These features characterize the stage of the awareness of the feelings: (1) The feeling of guilt is described by the participants as a permanent state that has been accompanying them since their childhood; (2) people whose parents were alcohol addicts cannot and, in some cases, do not want to use coping strategies focused on solving their problems. The participants described the feeling of guilt as an undifferentiated emotional state. For example, one of the participants said: “Everything is bad, I feel guilty every day”. The coping strategies of repression and suppression are used to overcome the feeling of guilt.

The stage of studying the characteristics of the feelings and their impact on their families is characterized by these features: (1) The feeling of guilt is described by the participants as a permanent state, but it may decrease or increase in some situations; (2) the participants understand the difference between the feeling of guilt and other emotions; (3) the participants can give specific examples of when they felt guilty.

The stage of the search for the resource factors of the patient’s family system is characterized by these features: (1) The participants of the program can give examples of the resource factors of their family systems; (2) the resource factors of the patient’s family system included keeping a diary, participation in a rehabilitation program, and confidential communication with other people; (3) the participants of the program pay attention to their own emotions and try to relate them to the situation in their family and the current state of the patient. As a result of participating in a rehabilitation program, in addition to avoiding and denying their problems, participants are able to formulate other coping strategies, such as focusing on emotions and their active expression in the rehabilitation program or searching for emotional support from friends and psychologists.

## 4. Discussion

The aim of this research was to investigate the coping strategies of people whose parents had alcohol addictions. The results showed that people whose parents had alcohol addictions have a tendency to avoid accepting responsibility for the difficulties of their families. People whose parents were alcohol addicts more often used escape–avoidance coping strategies than people from healthy families; the results showed a large effect between these groups (*p* < 0.001; *d* = 0.908). In the rehabilitation process, people whose parents had alcohol addictions learn to use a variety of coping strategies and to use emotional support from friends and psychologists. The obtained data correlate with the results of other research. J. Woititz wrote that adults whose parents suffered from alcohol addiction usually feel guilty and use only denial to lower these feelings, but these investigations were conducted on people who were chemical addicts [12]. In the current study, the experimental group consisted of mentally healthy adults who were not chemical addicts but whose parents suffered from alcohol addiction, but these people usually feel guilty too. 

The feeling of guilt was connected with the sense of responsibility. The results showed that the guilt of responsibility experienced by people whose parents had alcohol addictions scored significantly higher than the guilt of responsibility experienced by people from the control group (*p* < 0.05; *d* = 0.556). People whose parents had alcohol addiction felt guilty in situations where they tried to take care of someone, and they did not understand the process because their own parents did not model (and teach them) caretaking behavior. According to A. Freud, the feeling of guilt is formed at the age of five and is closely associated with the communication between the child and the parents [20]. A.V. Kuritsyn studied children whose parents had alcohol addiction, and wrote that these children often tended to apologize for their behavior and to take responsibility for their parents’ alcoholism [2]. A.V. Kuritsyn believed that this tendency was not observed in adulthood due to the effective coping strategies, but in our research, adult people felt the guilt of responsibility. 

We suggested that the feeling of guilt experienced by people with alcohol-addicted parents is associated with the actual condition of their parents. V.D. Moskalenko wrote that alcohol addiction is a family problem, and the feeling of guilt experienced by everybody in alcoholics’ families is associated with the actual condition of the alcohol addict [11]. However, in this research, the feeling of guilt experienced by adults whose parents had alcohol addictions was not associated with the actual condition of their parents. It was associated with the sense of responsibility, the fear of the death of their parents, and the feeling of anger. These feelings related to the pasts of the participants or concerned their fears of the future. These results may be useful for family system psychotherapy, which should focus on the present of the family and on the actual condition of the family. 

The feeling of guilt can be transformed during the process of participation in the twelve-step rehabilitation program “Adult Children of Alcoholics”. There are three main stages of overcoming the feeling of guilt experienced by people with alcohol-addicted parents in the rehabilitation process. At the end of the program, the participants could list the resource factors (keeping a diary, participation in a rehabilitation program, and confidential communication with other people) that they did not notice at the beginning of the program, and gave examples of how they used these resource factors in their own families. 

Some limitations are found within this study. The sample included only people who did not suffer from alcohol or other addictions and who were the participants of the twelve-step rehabilitation program for adults whose parents had alcohol addictions. We did not study people who were not participants of the twelve-step rehabilitation program. It would be useful to study these people because they do not want to use the participation in the rehabilitation program as a primary coping strategy. We expect that these people prefer to use coping strategies that do not involve other people—for example, exercising self-control, meditating, and positive reappraisal of their childhood. It would be useful to study people whose parents suffered from alcohol addiction and died versus people whose parents have alcohol addictions and live with their children. We expect that people whose parents were alcohol addicts and died feel guilty in the experience of the death of close relatives and do not feel guilt as a taboo offense against the parents. If this expectation is true, in the psychological counseling of people whose parents were alcohol addicts and died, it is necessary to work with the experience of death, and it is not useful to work with the feeling of anger. We expect that people whose parents have alcohol addictions and live with their children usually feel guilt as a taboo offense against the parents. If this expectation is true, in the psychological counseling of people whose parents have alcohol addictions and live with their children, it is important to train people whose parents have alcohol addictions to express their anger in a socially acceptable form. However, the purpose of this research was to investigate the feeling of guilt experienced by adults whose parents suffered from alcohol addiction and who were the participants of the twelve-step rehabilitation program for adults whose parents suffered from alcohol addiction. The results showed the main features of experiencing the feeling of guilt and ways out of it.

## 5. Summary 

We studied the feeling of guilt experienced by adults whose parents suffered from alcohol addiction and their coping strategies for lowering these feelings. People whose parents had alcohol addictions felt guilty in situations where they tried to take care of someone, and they did not understand the process because their own parents did not model (and teach them) caretaking behavior. Adults whose parents suffered from alcohol addiction tend to deny the difficulties of their families. In the rehabilitation process, these individuals learned to use a variety of coping strategies as well as the emotional support of friends and psychologists. As a result of participating in rehabilitation programs, in addition to avoiding and disclaiming their problems, participants were able to formulate other coping strategies, such as focusing on emotions and their active expression in the rehabilitation program or searching for emotional support from friends and psychologists. The feeling of guilt experienced by adults with alcohol-addicted parents was not associated with the actual condition of their parents; it was associated with the sense of responsibility, the fear of the death of their parents, and the feeling of anger. These feelings related to the pasts of the participants or concerned their fears of the future. These results may be useful for family system psychotherapy, which should focus on family communication and on the actual condition of the family. 

## 6. Conclusions

The coping strategies of the patients’ family systems included keeping a diary, participation in a rehabilitation program, and confidential communication with other people.

## Figures and Tables

**Table 1 behavsci-10-00032-t001:** Means, standard deviations, and the Kolmogorov–Smirnov statistic for each group compared for each variable.

Scales	M	SD	Z	*p*
The Guilt Inventory Questionnaire (GIQ), the state of guilt cont.	27.6400	7.97895	0.120	0.071
GIQ, the state of guilt ex.	27.9615	7.57954	0.094	0.200
GIQ, the feeling of guilt cont.	56.5800	14.68595	0.077	0.200
GIQ, the feeling of guilt ex.	63.9038	12.63400	0.132	0.025
GIQ, moral norms cont.	41.2000	7.24217	0.131	0.032
GIQ, moral norms ex.	40.3846	6.86038	0.174	0.000
GIQ, survivor guilt cont.	45.8400	9.19929	0.114	0.100
GIQ, survivor guilt ex.	48.6346	9.12233	0.079	0.200
GIQ, separation guilt cont.	38.4600	9.40302	0.085	0.200
GIQ, separation guilt ex.	34.4423	9.57409	0.080	0.200
GIQ, responsibility guilt cont.	61.4800	12.9461	0.064	0.200
GIQ, responsibility guilt ex.	68.8077	13.42815	0.099	0.200
GIQ, shame/self-hate guilt cont.	40.0400	13.71720	0.128	0.040
GIQ, shame/self-hate guilt ex.	44.4038	13.75736	0.106	0.200
Confrontational coping cont.	8.8200	3.02162	0.133	0.027
Confrontational coping ex.	9.6346	3.29604	0.083	0.200
Distance cont.	9.2400	3.83092	0.101	0.200
Distance ex.	8.8654	3.36084	0.094	0.200
Self-control cont.	10.0600	3.20974	0.095	0.200
Self-control ex.	10.7692	2.97450	0.142	0.010
Search for social support cont.	10.9200	2.70178	0.108	0.200
Search for social support ex.	11.0385	2.50460	0.147	0.007
Acceptance of responsibility cont.	7.2200	2.40993	0.144	0.012
Acceptance of responsibility ex.	7.0195	2.76172	0.132	0.025
Escape–avoidance cont.	10.7200	3.28286	0.086	0.200
Escape–avoidance ex.	14.0769	4.05282	0.072	0.200
Planning a solution to a problem cont.	10.9400	3.18421	0.101	0.200
Planning a solution to a problem ex.	9.9615	3.56982	0.113	0.098
Positive revaluation cont.	9.4200	3.51144	0.135	0.024
Positive revaluation ex.	9.5577	3.64295	0.106	0.200

cont.—control group, ex.—experimental group.

**Table 2 behavsci-10-00032-t002:** Coping strategies and the feeling of guilt experienced by people whose parents were alcohol addicts versus controls (parametric results, nonparametric results, and effect sizes).

Scales	*t*	*p*	U	*p*	Cohen’s d
The Guilt Inventory Questionnaire, the state of guilt	0.209	0.835	1,278,000	0.883	0.041
The Guilt Inventory Questionnaire, the feeling of guilt	2.703	0.008	950,000	0.019	0.535
The Guilt Inventory Questionnaire, moral norms	0.584	0.561	1,209,500	0.544	−0.116
GIQ, survivor guilt	1.540	0.127	1,113,000	0.210	0.305
GIQ, separation guilt	2.137	0.035	990,500	0.038	−0.423
GIQ, responsibility guilt	2.804	0.006	923,500	0.012	0.556
GIQ, shame/self-hate guilt	1.604	0.112	1,080,500	0.141	0.318
Confrontational coping	1.300	0.197	1,082,000	0.142	0.257
Distance	0.526	0.600	1,239,500	0.684	−0.104
Self-control	1.158	0.250	1,175,500	0.402	0.229
Search for social support	0.230	0.819	1,219,000	0.584	0.046
Acceptance of responsibility	0.391	0.697	1,278,000	0.882	−0.077
Escape–avoidance	4.586	0.000	674,000	0.000	0.908
Planning a solution to a problem	1.459	0.148	1,102,000	0.183	−0.289
Positive revaluation	0.194	0.846	1,273,000	0.856	0.038

## Data Availability

The datasets used and analyzed during the current study are available from the corresponding author on reasonable request.

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
