# Peer review of "Coping Strategies and Personality Profile Characteristics of People Whose Parents Were Alcohol Addicts"

_behavsci, 2020, doi:10.3390/bs10010032_

Round 1

Reviewer 1 Report

I am reviewing the paper entitled “Coping Strategies and Personality Profile Characteristics of People Whose Parents Were Alcohol Addicts” submitted to Behavioral Sciences, and the paper is a fairly straightforward article depicting the benefits of a 12-step intervention program for adults whose parents where alcohol addicts compared to controls. The authors should fix or change some graphs and I see some writing problems that should be improved, and I will describe these issues in detail.

The Figure on page 3 is super confusing. What are the authors presenting? I realize that it should be something to due with a guilt questionnaire, but the reader needs more information. Specifically, a figure needs to label the y-axis and the x-axis. The authors present 49 items on the x-axis. Therefore, I am inferring that they presented the information for every participant? Why? Journals do not have room for such individualized presentations.   The authors should present a bar graph showing the means for the control and experimental groups. The bars should also be different enough (color or some other way to discriminate) to distinguish them from each other. On page 4, the author presents a pie graph. Businesses love pie graphs because they are pretty, but the primary reason to present information graphically is to get away from numbers, but pie graphs do not present a uniform starting place, so they need numbers to help the reader differentiate the different slices, so bar graphs defeat the purpose of using figures. Bar graphs work much better to put all the groups in the same starting position and visual comparisons can easily be made. In addition, the slices in the pie graph cannot really be distinguished due to the similar coloring used. Therefore, the bars should easily discriminate between the different bars.

All the phenomenological information presented in section 3.2 should be presented in an Appendix. The authors can certainly interpret the results in that section and provide a few select statements to make necessary and important points.

On line 15 in the abstract, the authors should remove the colon after “We used” and they should say “the COPE questionnaire” on line 17 and the authors say “a lot of qualitative methods” and they should improve their language there. On line 19, the authors should say “The results showed”. On line 20, the authors say that adults with alcohol addict parents felt guilty, but the reasoning is wrong. These individuals did not feel guilty because their parents did not take care of them. Rather, these individuals felt guilty because their own parents did not model (and teach them) caretaking behavior because their parents did not take care of them. On line 21, the authors say “addicts have a tendency” but they could simply say “addicts tend to disclaim”.

The authors should put a semicolon after “of the patient” on line 32. Line 33 says “a lot of family factors”. The authors can improve upon this statement. Line 37 starts off in a confusing manner saying “In the family system psychotherapy”. I am not sure what the authors are saying, so they should be clear and I am pretty sure that a comma should follow this sentence fragment. Line 39 should delete the “the”s in front of homeostasis and development. Line 46 should say “the feeling of guilt”. Line 49 uses “he” and “his” and most current writing attempts to make the pronouns gender neutral “s/he” or “their” (I personally do not like making something plural that is singular) but “her” would be preferable to “he” and “his”. On line 54, the authors could say “These people do not understand “normal” family behavior, they report low self-esteem, they frequently experience guilt and difficulty in establishing intimate relationships in their own families, and they rarely seek new employment opportunities.” I am guessing on the last point (was the authors’ second point about work) because I had no idea what the authors were trying to convey. In the next sentence, the authors should say “In the Russian studies conducted by …, children in alcohol families tended to be co-dependent, they demonstrated addictive and suicidal behavior, they reported low self-esteem and they experienced difficulties in establishing…” On line 61, the authors use “A lot of research”. The phrase “a lot of” is simply too colloquial. Line 64 should say “In the current study, the experimental group”. Why don’t the authors describe the control group at this point? The authors do not need to capitalize “The” on line 71.

Line 79 should say “Inclusion criteria required all participants to range from…age, report being mentally healthy”. Line 82 says “his life”. Line 83 places a large gap between words. Line 84 should say “5 years old”. Line 92 should not put a colon after “We used” and it should follow the comments made for the abstract. Line 97 should say “The results for the guilt scale primarily pertained to three subscales”. After listing the first two scales, the authors should put an “and” before “moral norms”. Line 99 uses “he” again and Line 100 should talk about the third scale, which is moral norms. Lines 101-104 should change the language to sound like the language suggested for line 97. Lines 120-129 should present means, t-test values, and then p-values. That authors do not need to present means for means presented in figures. The authors should not say “Picture 1” in Figure 1.

Line 151 should put a comma after “5”. Lines 195-196 should remove the three instances of “the stage of”. The same is true on line 198. Lines 198-203 make for an incredibly long sentence. I am also confused about the point the authors are trying to make in this paragraph/sentence. Line 230 uses “this” but this what? The authors should probably say “when they tried to take care of someone, and they did not understand the process because their parents did not model this behavior because their parents did not take care of them.” Line 233 should delete “had a tendency”. Line 248 should put a comma after “program” and then use past tense when talking about giving examples and did the participants provide examples or how was this ability tested? Line 255 talks about coping strategies, but which ones? Provide examples. For lines 255-257, why would this line of research be interesting? Which questions would it possibly answer? What do the authors expect?

Conclusion should be used for the last sentence of the summary section. The authors should use “Summary” instead of “Conclusions”. Line 263 should say feelings. Line 264 could use more formal language as suggested previously. Line 265 should remove “have a tendency” and simply say “tend to” Line 266 should say “these individuals learned”. Line 267 should say “coping strategies as well as the emotional”. The next sentence should start “As a result”; capitalize “as”. Line 269 should say “were” instead of “are” and put a comma after “coping strategies”. Line 272 should put a semicolon after “parents” and start the next fragment as “it was”. Line 275 says “on the present of the family” and this phrasing is very confusing. Line 277 should put a comma after “rehabilitation program”.

Author Response

Dear Reviewer!

Thank you very much for your review, it was very informative for me.

I deleted the graphs and added 2 tables because my graphs were not informative and the other reviewer asked me to add several data to my article. 

All the phenomenological information presented in section 3.2 is presented in an Appendix. 

I changed all writing problems that you described and you can see it in the new file.

Best regards, Anna Lutsenko. 

Reviewer 2 Report

1.  There are spacing problems where large gaps were left between words.

2.  Line 64.  I'd suggest "In this study the experimental group"

3.  Under 3.1, please provide standard deviations as well as means. 

4.  Line 97  "a lot of" isn't good scientific language; try "several qualitative methods" instead

5.  Line 111   I'd recommend "statements, and grouping the"

6.  118  While using nonparametric statistics is sometimes helpful, it would be useful to know if the variable distributions were non-normal or not.  Even if so, I'd prefer having both parametric and nonparametric results presented, including means and standard deviations for each group compared for each variable compared.  Otherwise, it's difficult to know what was really happening with the data.  Effect sizes should also be reported (Cohen's d).

7.  The word "disclaim" is used frequently but it's English meaning is unclear in this context.  What is meant?  Deny?  Minimize?  "Avoid accepting responsibility for"?  Please clarify.

Author Response

Dear Reviewer!

Thank you very much for your review, it was very informative for me.

I deleted the large gaps between words. I changed "In the current study, the experimental group" I provided standard deviations and means. I deleted this phrase because it was not informative and provided only methods. I changed to "statements, and grouping the" I did it, you can see all results (means and standard deviations for each group compared for each variable compared, both parametric and nonparametric results and effect sizes) (Table 1 and Table 2). I mean "Avoid accepting responsibility for" and I clarified this in my article in abstract, discussion and summary.

I changed some writing problems according to other review report.

I submit you the new version of my article. 

Best regards, Anna Lutsenko 

Round 2

Reviewer 2 Report

1.  line 12 i'd suggest "resource factors and coping strategies"

2.  line 14  I'd suggest "52 subjects, who were participants"

3.  line 18  I'd suggest ", and the phenomenological analysis"

4.  line 29  I'd suggest "with the very wide prevalence"

5.  The major flaw in this paper at present is the apparently incorrect t-test values in Table 2.  For example, for The Guilt Inventory feeling of guilt, the t is shown as 0.185 but my calculation finds the t value to be 2.70; the significance levels are correct (i.e., .008) it's just the t value seems incorrect.  For escape-avoidance, the t value I calculate is 4.59 with p < .001 but the paper has t of .131.  It appears that most of the reported t-values are incorrect and must be revised.  If that is not done, the paper should not be published.

6.  For the right hand column in Table 2 I would suggest "Cohen's d" rather than d'Cohen. 

Author Response

Dear Reviewer!

Thank you very much for your review.

1)  line 12 I  changed to "resource factors and coping strategies" 

2) line 14 I  changed to "52 subjects, who were participants"

3)  line 18 I changed to ", and the phenomenological analysis"

4)  line 18 I changed to ", and the phenomenological analysis"

5) I revised all  t-test values in Table 2 according your recommendations. Thank you very much for pointing me to this flaw, I usually work with qualitative methods. In the first version of my article I accidentally indicated the Levene's test values instead of the t-test values, but in the new version of my article I revised all t-test values. 

6) For the right hand column in Table 2 I changed to "Cohen's d" 

Best regards, Anna Lutsenko

Round 3

Reviewer 2 Report

You might enrich the discussion by taking the meaning of Cohen's d into account.  Cohen (1992) in Psychological Bulletin with an article entitled "A power primer" described .20 as a small effect, .50 as a medium effect, and .80 as a large effect.  The .50 effect size was enough that a careful non-trained observer would probably notice the difference, a "naked eye" effect, so to speak.  The relative importance of the findings in the t-tests can be based on the size of the Cohen's d's.

Author Response

Dear Reviewer!

Thank you for your review.

I added the Cohen's d results in the results, and took the meaning of Cohen's d into account in the discussion.

Best regards, Anna Lutsenko